# LK Losses: Direct Acceptance Rate Optimization for Speculative Decoding

**Alexander Samarin** [1]  **Sergei Krutikov** [1]  **Anton Shevtsov** [1]  **Sergei Skvortsov** [1]  **Filipp Fisin** [1]  **Alexander Golubev** [1]

## Abstract

Speculative decoding accelerates autoregressive large language model (LLM) inference by using a lightweight draft model to propose candidate tokens that are then verified in parallel by the target model. The speedup is significantly determined by the acceptance rate, yet standard training minimizes Kullback-Leibler (KL) divergence as a proxy objective. While KL divergence and acceptance rate share the same global optimum, small draft models, having limited capacity, typically converge to suboptimal solutions where minimizing KL does not guarantee maximizing acceptance rate. To address this issue, we propose **LK losses**, special training objectives that directly target acceptance rate. Comprehensive experiments across four draft architectures and six target models, ranging from 8B to 685B parameters, demonstrate consistent improvements in acceptance metrics across all configurations compared to the standard KL-based training. We evaluate our approach on general, coding and math domains and report gains of up to 8-10% in average acceptance length. LK losses are easy to implement, introduce no computational overhead and can be directly integrated into any existing speculator training framework, making them a compelling alternative to the existing draft training objectives.

## 1. Introduction

Large language model (LLM) inference is fundamentally constrained by memory bandwidth rather than computational throughput. The autoregressive nature of token generation requires sequential memory accesses that underutilize modern accelerators, creating a critical bottleneck for deployment at scale. Speculative decoding (Leviathan et al.,

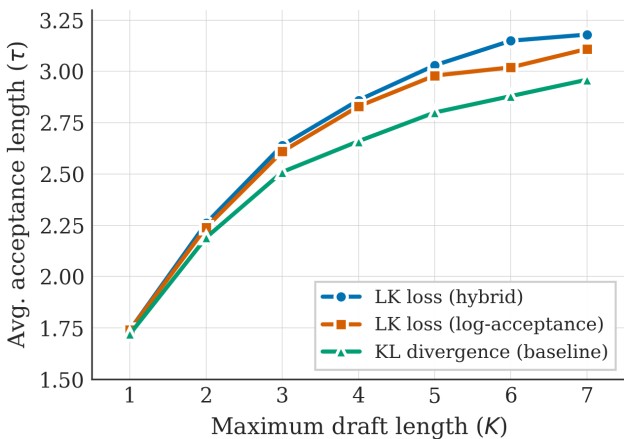

*Figure 1.* Acceptance length $\tau$ vs maximum length $K$ for EAGLE-3 draft models, trained using different objectives with Qwen3-235B-A22B-Instruct as a target model. The values were obtained on the MT-Bench dataset with chain sampling at temperature = 1.

2023; Chen et al., 2023) addresses this challenge through a draft-then-verify paradigm: a lightweight draft model proposes multiple candidate tokens, which the target model then verifies in a single forward pass. This approach preserves the output distribution of the target model while achieving substantial speedups. The efficiency of this process is largely determined by the *acceptance rate* – an expected probability of a drafted token being accepted by the target model.

A variety of draft model architectures implementing that idea have emerged, including parallel prediction heads (Cai et al., 2024; Sandler et al., 2025), autoregressive draft heads with feature fusion (Li et al., 2024; 2025b), and multi-token prediction modules that can be repurposed for speculation after training (DeepSeek-AI et al., 2024). These approaches commonly train draft models by minimizing Kullback-Leibler (KL) divergence between target and draft distributions, treating distributional alignment as a proxy for acceptance rate optimization. At the global optimum draft matches target perfectly, so this proxy is exact: KL divergence reaches zero and acceptance rate reaches one. However, draft models operate under severe capacity constraints, typically having 1-5% of the target model parameters, and

[1]Nebius, Amsterdam, Netherlands. Correspondence to: Alexander Golubev <alex_golubev@nebius.com>.

*Proceedings of the 43rd International Conference on Machine Learning*, Seoul, South Korea. PMLR 306, 2026. Copyright 2026 by the author(s).

inevitably converge to suboptimal solutions. At these suboptimal points, minimizing KL divergence provides no guarantee of maximizing acceptance rate. Leviathan et al. (2023) noted that greater improvements might be obtained via custom training procedures, including direct optimization of the draft model for the acceptance rate, yet this direction has remained largely unexplored.

In this work, we propose two variants of **LK losses** [1], training objectives that directly target acceptance rate. One of them is motivated by the maximum likelihood methods, directly optimizing the negative log-acceptance rate. The other variant is a hybrid approach, which gradually shifts the focus from KL towards direct acceptance optimization as training progresses, analogous to trust-region methods that balance a stable surrogate objective against the true optimization target. Both losses help improving average acceptance length, especially for longer draft sequences (see Figure 1).

In summary, our contributions are as follows:

- We propose two loss variants for direct acceptance rate optimization in speculative decoding.

- We demonstrate consistent improvements in acceptance metrics across a number of target models and draft architectures, empirically confirming that LK losses are both model- and architecture-agnostic.

- We release our training datasets and draft model weights to facilitate reproducibility and further research [2].

## 2. Related Work

### 2.1. Draft Model Architectures

Early works on speculative decoding utilized smaller, standalone versions of the target model as draft models (Leviathan et al., 2023; Chen et al., 2023). Although straightforward to implement, such an approach usually suffers from performance bottlenecks within the draft model. Moreover, to achieve high acceptance rate it either needs a small pretrained language model (LM) from the same family or training from scratch on a sufficiently large corpus of data.

To mitigate these overheads, subsequent works integrated the drafting mechanism directly into the target model.

MEDUSA (Cai et al., 2024) attaches parallel decoding heads to the target model's final layer to predict draft tokens independently from each other. This approach is computationally efficient but implicitly assumes conditional independence between the proposed tokens, which can degrade performance for distant draft positions.

Other methods introduce autoregressive draft heads that also operate in the target model's hidden state space. Wertheimer et al. (2024) propose a multi-stage MLP speculator that extends MEDUSA heads with the principles from recurrent networks. A more advanced EAGLE (Li et al., 2024; 2025b) family employs a shallow transformer model with a causal mask to better capture long-term dependencies between draft tokens and their context. EAGLE-3 also enriches the input feature space of the speculator heads by fusing hidden states from various intermediate layers of the target model.

A similar autoregressive design was recently adopted by DeepSeek-V3 with its Multi-Token Prediction (MTP) module (DeepSeek-AI et al., 2024). It predicts multiple draft tokens during training, essentially serving as a native "draft head" that requires no separate post-training.

In addition to architectural advances, some works address the problem of computational overhead in the LM heads of the draft models. FR-Spec (Zhao et al., 2025) observed that large token vocabularies of contemporary LLMs often make the latency of LM heads dominate in the total latency of the speculator heads, even in such powerful approaches as EAGLE. FR-Spec addresses this issue by truncating the draft vocabulary to a small subset of high-frequency tokens learned on training data.

### 2.2. Training Methodologies for Draft Models

**Knowledge Distillation.** The problem of training speculators has been predominantly framed as the Knowledge Distillation (KD). Within this paradigm, draft models are students that approximate the teacher's (i.e. target) output distribution as closely as possible. The standard training objective has therefore been KL divergence, or equivalently cross-entropy (CE), between the target and draft distributions.

Some works experimented with combined objectives. MEDUSA suggests mixing the KL loss with the LM objective for the target model to address the discrepancy between its distribution and static training data. EAGLE (Li et al., 2024) extends KL with regression loss on hidden states of the last layer in an attempt to better match training and inference settings.

**Targeting Acceptance Rate.** Some studies went beyond the standard KD framework in terms of both training data and objectives. DistillSpec (Zhou et al., 2024) explores other types of divergences, including reverse KL and total varia-

---

[1] The naming LK positions the loss as an alternative to KL divergence: standard approaches minimize KL as a proxy for acceptance rate, while LK objectives target acceptance directly. The notation also connects to the $D_{LK} \equiv$ TV divergence in the original speculative decoding paper (Leviathan et al., 2023).

[2] Datasets: HuggingFace nebius/infinity-instruct-completions, Weights: HuggingFace nebius/lk-speculators.

tion (TV) distance, as KD objectives for already pretrained LMs being used as external speculators. Although they note that TV distance should theoretically be the right objective as it directly maximizes acceptance rate, they conclude that the choice of the divergence loss is highly dependent on the task and data being used for KD. AdaSPEC (Hu et al., 2025) further addresses the mismatch between standard KD objectives and speculative decoding efficiency through selective distillation strategies for standalone draft models. In contrast, our work focuses on native draft modules tightly integrated into target models and trained from scratch, rather than on separately pretrained external drafters. Within this setting, we study whether the core training objective should optimize token acceptance directly instead of proxy divergence measures.

## 3. Background and Motivation

### 3.1. Speculative Decoding

In this paper we are working under the standard lossless speculative sampling setting. A draft model $q$ proposes a sequence of tokens $x_1, \ldots, x_K$ given the context $\mathbf{c}$ that are later verified by the target model $p$ in parallel. Each draft token $x_i$ in the sequence is accepted with the probability

$$\beta(x_i \mid \mathbf{c}, \mathbf{x}_{<i}) = \min\left(1, \frac{p(x_i \mid \mathbf{c}, \mathbf{x}_{<i})}{q(x_i \mid \mathbf{c}, \mathbf{x}_{<i})}\right),$$

where $\mathbf{x}_{<i} = (x_1, \ldots x_{i-1})$ is a prefix of the $i$-th draft token. Drafted tokens are verified in parallel and accepted sequentially. The first rejected token terminates the accepted sequence, discarding all subsequent drafts.

The efficiency of speculative decoding is mainly driven by the *acceptance rate* defined as an expected acceptance probability for the $i$-th draft token

$$\begin{aligned}
\alpha_i &= \mathbb{E}_{x \sim q(\cdot \mid \mathbf{c}, \mathbf{x}_{<i})}\left[\beta(x \mid \mathbf{c}, \mathbf{x}_{<i})\right] \\
&= \sum_{x \in \mathcal{V}} \min\left(q(x \mid \mathbf{c}, \mathbf{x}_{<i}), p(x \mid \mathbf{c}, \mathbf{x}_{<i})\right),
\end{aligned} \quad (1)$$

where $\mathcal{V}$ is the vocabulary of tokens. In the following sections we omit the conditioning and subscripts for brevity wherever the context is clear. From the definition of $\alpha$ above it follows that its global optimum is achieved at $q = p$.

### 3.2. Divergence Losses

Statistical divergences are defined as discrepancy measures between probability distributions (Amari & Nagaoka, 2000). As it is noted in Section 2.2, they are widely employed as primary training objectives for draft models. The most commonly used divergences (Zhou et al., 2024) include **forward KL divergence**,

$$\mathrm{KL}(p\|q) = \sum_i p_i \log(p_i/q_i),$$

**reverse KL divergence** $\mathrm{KL}(q\|p)$ and **Total Variation distance**

$$\mathrm{TV}(p, q) = \frac{1}{2} \sum_i |p_i - q_i|.$$

The TV distance is of great interest for us due to its direct relation to acceptance rate via $\alpha = 1 - \mathrm{TV}(p, q)$ (Leviathan et al., 2023). Thus, maximization of acceptance is strictly equivalent to minimization of TV whereas other divergences serve only as proxy objectives. The gap between proxy and direct optimization becomes stark when model capacity is limited, which is demonstrated in Section 3.3.

Direct influence of TV on acceptance has been noticed and employed in prior works (Zhou et al., 2024; Yin et al., 2024), providing competitive results compared to other training objectives. However, in our setting draft model is initialized randomly rather than from a pretrained language model. We reveal why this distinction is critical in Section 4.1.

### 3.3. Motivating Example

An important property of the TV distance is its focus on the tokens that constitute the major probability mass under the target distribution (Ji et al., 2023). This is a major advantage given the limited capacity of the draft model and its inability to fully match the target distribution. Figure 2 illustrates this with a simple example – fitting a single Gaussian to a multi-modal Gaussian mixture distribution using different divergences as loss functions. Green areas visualize density overlap, which in fact is equal to the acceptance rate if we were applying speculative sampling algorithm to this toy example (see Appendix C).

Forward KL divergence, being mode-covering, spreads probability mass broadly to avoid infinite penalties wherever the target has support, resulting in suboptimal acceptance rate. Reverse KL divergence exhibits the opposite failure mode, underestimating uncertainty and collapsing to cover only the dominant mode. In contrast, TV distance finds a qualitatively different solution that maximizes the distributional overlap, achieving substantially higher acceptance despite using the same parametric family. This toy example highlights an important observation: when the draft cannot perfectly match the target, the choice of objective determines which compromises the optimization makes.

## 4. Methodology

We propose training objectives that directly target acceptance rate rather than using KL divergence as a proxy. Our approach is grounded in gradient analysis that reveals fundamental differences in how various divergence measures guide optimization.

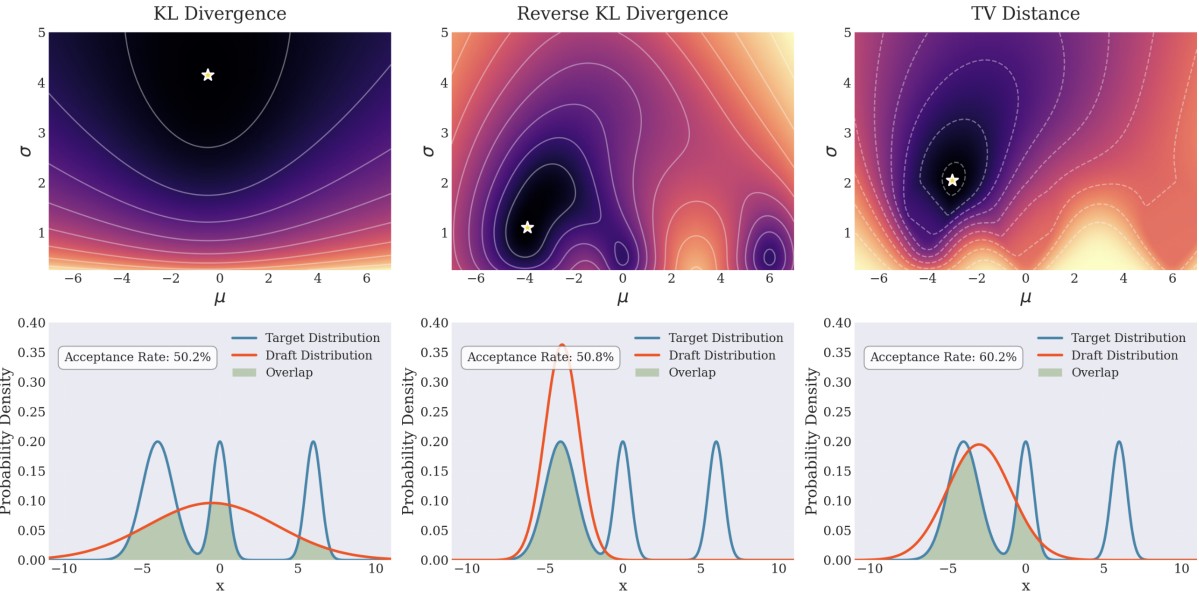

*Figure 2.* Fitting a single Gaussian to a Gaussian mixture under different objectives. **Top:** Loss landscapes (log-scale) over parameters $\mu$ and $\sigma$. **Bottom:** Resulting distributions and overlap (green). KL divergence produces a mass-covering solution ($\alpha = 50.2\%$), reverse KL exhibits mode-seeking behavior ($\alpha = 50.8\%$), while TV maximizes overlap ($\alpha = 60.2\%$).

### 4.1. Gradient Analysis of Divergence Losses

Consider training a parametric draft model with logits $z_q$ to match a target distribution $p$ through minimization of divergence loss, where $q = \mathrm{softmax}(z_q)$. The choice of the divergence type fundamentally determines optimization dynamics through its gradient structure. Understanding that is especially important in case when training starts with randomly initialized parameters, which implies large discrepancy between the draft and target distributions. In this section we explore two most relevant divergence losses, forward KL and TV.

The forward KL divergence yields an elegant gradient with respect to the logits (see A.2 for derivations):

$$\nabla_{z_q} \mathrm{KL}(p\|q) = q - p. \qquad (2)$$

This gradient pushes each logit proportionally to the gap between predicted and target probabilities. At the early stage of the training the gradient magnitude $\|q - p\|$ scales as $\mathcal{O}(1/\sqrt{k})$ when $p$ is concentrated on $k$ tokens (see A.5), providing strong signal regardless of current alignment.

The gradient of TV distance takes more sophisticated form (see A.3)

$$\nabla_{z_q} \mathrm{TV}(p,q) = \frac{1}{2} q \odot \left(s - \mathbb{E}_q[s]\right), \qquad (3)$$

where $s_i = \mathrm{sign}(q_i - p_i)$ and $\mathbb{E}_q[s] = \sum_i q_i s_i$. While this gradient provides directional information about whether

each token is under- or over-predicted, the signal depends only on the sign of the error, not its magnitude. A token with $q_i$ slightly below $p_i$ receives the same sign signal as one severely under-predicted, which contrasts with KL divergence gradient. The TV loss landscape also contains non-differentiable points along the manifold $\{z_q : q_i = p_i\}$, where gradients change discontinuously.

A more significant concern is gradient magnitude. For randomly initialized draft models which spread $q$ over a large vocabulary of size $V$, the gradient norm satisfies $\|\nabla_{z_q} \mathrm{TV}\| = \mathcal{O}(\sqrt{k}/V)$ (see A.5). With typical vocabulary sizes exceeding 100k, this yields extremely small gradients at initialization. Together with ignorance of error magnitude and the non-smooth loss landscape, these issues make pure TV optimization impractical for training from random initialization.

### 4.2. Hybrid Objective with Adaptive Blending

The gradient analysis reveals a fundamental controversy. KL divergence creates smooth, well-conditioned optimization landscapes but optimizes a *proxy* for the acceptance rate. Conversely, directly minimizing TV distance targets the correct objective but suffers from vanishing gradients and non-smooth optimization surfaces. To benefit from both advantages, we propose a hybrid objective that combines KL divergence with TV distance as follows:

$$\mathcal{L}_{\mathrm{LK}}^{\lambda}(p,q) = \lambda \cdot \mathrm{KL}(p\|q) + (1 - \lambda) \cdot \mathrm{TV}(p,q). \qquad (4)$$

With $\lambda = 1$ we recover standard KL training whilst setting $\lambda = 0$ leads to pure TV optimization.

The key insight is that these components serve complementary roles at different training stages. Early in training, when $q$ is far from $p$, the KL component provides smooth, properly-scaled gradients that efficiently navigate the loss landscape. As alignment improves, the TV component takes over to directly optimize acceptance rate.

**Adaptive schedule.** To achieve the aforementioned behavior, we propose the following schedule driven by the ongoing acceptance rate value:

$$\lambda = \exp\left(-\eta \cdot \mathrm{sg}[\alpha]\right), \quad \eta > 0, \tag{5}$$

where $\mathrm{sg}[\cdot]$ denotes stop-gradient operation, preventing backpropagation through $\lambda$. We compute $\lambda$ independently for each draft token position using aggregated values of $\alpha$ across sequence and batch dimensions.

The proposed schedule satisfies the desired properties: $\lambda \to 1$ when $\alpha \to 0$ (poor alignment), and $\lambda$ converges to a small value when $\alpha \to 1$ (good alignment), smoothly transitioning from KL-dominated to TV-dominated optimization as training progresses. In Section 6 we empirically confirm through ablation studies that such an adaptive schedule makes the hybrid objective superior to pure TV and KL losses as well as to the fixed mixture of weights.

The hybrid objective with adaptive scheduling can be interpreted through the lens of constrained optimization. When $\alpha$ is small, large $\lambda$ prioritizes KL minimization, establishing a region where the draft distribution $q$ is sufficiently close to $p$ for TV gradients to behave well. As $\alpha$ increases and $\lambda$ decays, the objective shifts towards TV minimization while the KL term acts as a soft constraint, maintaining distributional proximity. This resembles the trust-region approach used in policy optimization (Schulman et al., 2015):

$$\min_q \mathrm{TV}(p, q) \quad \text{s.t.} \quad \mathrm{KL}(p\|q) \leq \delta,$$

where the adaptive schedule implicitly controls the effective constraint threshold $\delta$ based on current alignment quality.

### 4.3. Likelihood-based Approach

A different interpretation of the acceptance rate comes from its role in the speculative sampling algorithm highlighted in Section 3.1. By definition, $\beta(x)$ is the conditional probability of acceptance, given that token $x$ was drafted, and $q(x)$ is the probability of drafting token $x$. Thus, we can interpret

$$\alpha = \sum_{x \in \mathcal{V}} q(x)\, \beta(x).$$

as the marginal probability of acceptance. Therefore, it is natural to consider minimization of the negative log

marginal likelihood as a training objective that clearly maximizes $\alpha$, or more precisely

$$\begin{aligned}
\mathcal{L}_{\mathrm{LK}}^{\alpha}(p, q) &= -\log \alpha \\
&= -\log \sum_{x \in \mathcal{V}} \min(p(x), q(x)).
\end{aligned}$$

This objective looks appealing due to its simplicity, as it does not need a complex mixture of different losses with adaptive weight scheduling to tackle optimization challenges.

In the degenerate case where $p$ is a point mass, $\mathcal{L}_{\mathrm{LK}}^{\alpha}$ reduces to the standard negative log-likelihood (see Appendix B for details).

**Gradient behaviour.** We derive in A.4 that

$$\nabla_{z_q} \mathcal{L}_{\mathrm{LK}}^{\alpha} = \frac{1}{\alpha} \nabla_{z_q} \mathrm{TV}(p, q), \tag{6}$$

which reveals a key insight – $\mathcal{L}_{\mathrm{LK}}^{\alpha}$ performs TV optimization *with adaptive gradient scaling*. The $1/\alpha$ factor provides automatic amplification when acceptance is low ($\alpha \to 0$), addressing the vanishing gradient problem. The gradient magnitude $\|\nabla_{z_q} \mathcal{L}_{\mathrm{LK}}^{\alpha}\|$ matches the one of KL at the early stage of the training (see A.5), while the gradient direction matches that of TV.

We evaluate both $\mathcal{L}_{\mathrm{LK}}^{\alpha}$ and $\mathcal{L}_{\mathrm{LK}}^{\lambda}$ empirically in Section 6, finding that both improve over pure TV and KL losses, with the hybrid objective generally achieving stronger results.

### 4.4. Vocabulary Truncation

EAGLE-3 uses a truncated vocabulary for its LM head as it is proposed in FR-Spec (Zhao et al., 2025), which sets a large subset of draft probabilities to zero. This creates a fundamental problem in KL-based training: KL divergence becomes infinite for tokens outside the draft vocabulary as $q_i = 0$ and $p_i > 0$. To overcome this challenge, we redefine standard target probabilities $p = \mathrm{softmax}(z_p)$, where $z_p$ are the target model logits, as $\tilde{p} = \mathrm{softmax}(m \odot z_p)$, where the mask $m$ sets logits of tokens outside the draft vocabulary to $-\infty$. However, this introduces another layer of approximation as now we optimize $\mathrm{KL}(\tilde{p}\|q)$ rather than $\mathrm{KL}(p\|q)$, making KL a proxy of a proxy.

In contrast, LK losses handle vocabulary truncation naturally. From (1) it follows that tokens outside the draft vocabulary contribute $\min(p_i, 0) = 0$ to the acceptance rate and therefore have no influence. Thus, no modification of $p$ is needed for $\mathcal{L}_{\mathrm{LK}}^{\alpha}$ loss and TV component in $\mathcal{L}_{\mathrm{LK}}^{\lambda}$ which optimize acceptance rate with respect to the *original* target distribution rather than its approximation.

## 5. Experimental Settings

We evaluate LK losses across a diverse range of target models and draft architectures to assess their generality and

practical impact.

## 5.1. Target Models

Our experiments span six target models covering three orders of magnitude in parameter count. We include both dense models, namely Llama-3.1-8B-Instruct (Grattafiori et al., 2024) and Llama-3.3-70B-Instruct, and mixture-of-experts (MoE) models, namely gpt-oss-20b, gpt-oss-120b, Qwen3-235B-A22B-Instruct (Yang et al., 2025) and DeepSeek-V3 (DeepSeek-AI et al., 2024). This selection allows us to test whether the benefits of LK losses extend to different model scales and architectures.

## 5.2. Draft Models

We evaluate four speculator architectures which are most commonly present in industrial applications. For Llama-3.1-8B we train three architectures to enable direct comparison: EAGLE-3 (Li et al., 2025b), multi-stage MLP speculator (Wertheimer et al., 2024) and MEDUSA (Cai et al., 2024). For larger models (Llama-3.3-70B, gpt-oss, Qwen3), we train only EAGLE-3 with *dense* transformer block as the best-performing state-of-the-art architecture. For DeepSeek-V3, we fine-tune the native MTP module.

All draft model architectures are trained with $K = 6$ speculative heads. EAGLE-3 shares weights across positions via recurrence, while the MLP speculator and MEDUSA use fully independent heads at each position. The MTP module retains its original MoE architecture and is initialized from the pretrained DeepSeek-V3 weights.

**Rationale for MTP fine-tuning.** While our method is primarily designed for training draft heads from scratch, fine-tuning pretrained MTP modules addresses a complementary challenge. Released open-source weights include only the first MTP module, which was originally trained to predict the first token, yet reused autoregressively for later ones. (Liu et al., 2026; Cai et al., 2025). This mismatch causes sharp decline in acceptance rate across later positions. Our adaptive $\lambda$ scheduler naturally addresses this inconsistency. Early heads with fairly high acceptance receive low $\lambda$, focusing on TV-driven adjustments within the trust region to further improve acceptance. Conversely, later heads with degraded acceptance receive higher $\lambda$, providing stronger KL guidance precisely where the MTP module was not explicitly trained.

**Output vocabulary.** EAGLE-3 draft models include a trainable unembedding matrix for next-token prediction. We adopt truncated vocabularies from RedHatAI speculators,[3] using only the vocabulary definitions while training all the weights from scratch. For DeepSeek-MTP, we retain the

original full vocabulary to preserve compatibility with the pretrained module.

## 5.3. Training Configuration

We construct the training corpus using 660K prompts from Infinity-Instruct-0625 (Li et al., 2025a) and generating responses with each target model listed above. This ensures the draft model is trained on the same distribution it will encounter during inference. All models are trained with the input sequence length of 8K.

We use batch size of 64 and learning rate $4 \times 10^{-4}$ with cosine scheduling and 100 warmup steps. Following the original speculative decoding literature, we use AdamW optimizer with $(\beta_1, \beta_2) = (0.9, 0.95)$ and gradient clipping at 0.5. All draft models except DeepSeek-MTP are trained from scratch for 10 epochs. For DeepSeek-V3, we initialize from the original MTP weights and fine-tune for 1 epoch. We set temperature $T = 1$ to match our primary evaluation setting.

We compare the following loss configurations: forward KL divergence $\mathrm{KL}(p \| q)$, negative log-acceptance loss $\mathcal{L}_{\mathrm{LK}}^{\alpha}$ (Section 4.3) and hybrid objective $\mathcal{L}_{\mathrm{LK}}^{\lambda}$ (Section 4.2) with the adaptive scheduler and $\eta = 3$ unless stated otherwise.[4] For Llama-3.1-8B with EAGLE-3 we additionally explore total variation distance $\mathrm{TV}(p, q)$, $\mathcal{L}_{\mathrm{LK}}^{\lambda}$ with the fixed weight $\lambda = 0.5$ and adaptive $\mathcal{L}_{\mathrm{LK}}^{\lambda}$ with $\eta = 0.7, 1$ and 10.

All the losses are aggregated across draft heads with exponential weight decay $\gamma \in (0, 1]$, i.e., the $n$-th head receives weight $\gamma^{n-1}$, prioritizing early positions which have the largest impact on average acceptance length. In our experiments we set $\gamma = 0.8$, following MEDUSA and EAGLE configurations.

## 5.4. Evaluation Protocol

**Inference Framework.** We evaluate draft models using vLLM v0.11.0 (Kwon et al., 2023) with a patch that enables proper rejection sampling at non-zero temperatures.[5] In the original version, tokens are sampled greedily from the draft distribution regardless of temperature settings. Our modification implements a theoretically correct rejection sampling procedure from Leviathan et al. (2023). See Appendix D for a detailed analysis of how greedy draft sampling affects acceptance rates.

**Evaluation Datasets.** We measure acceptance metrics

---

[4]For MEDUSA we use $\eta = 10$ as its acceptance rates improve slower during training compared to recurrent architectures. Larger $\eta$ accelerates transition towards TV optimization to compensate.

[5]Our patch builds upon https://github.com/vllm-project/vllm/pull/20459. Equivalent stochastic sampling support was later integrated into upstream vLLM starting from v0.18.0.

on three benchmarks: multi-turn conversational prompts **MT-Bench** (Zheng et al., 2023), code generation tasks **HumanEval** (Chen et al., 2021), and grade-school math problems **GSM8K** (Cobbe et al., 2021). Unlike many prior works that evaluate draft models on small subsets of prompts (e.g. 80 samples in Li et al., 2025b), we evaluate on full datasets to obtain more reliable estimates of the acceptance metrics.

**Sampling Configurations.** We perform evaluations under two temperature settings: greedy decoding ($T = 0$) and stochastic sampling ($T = 1$). Our training objective directly optimizes acceptance probabilities under stochastic sampling, making it the primary evaluation setting. We additionally report greedy decoding results to demonstrate generalization across different sampling regimes.

We evaluate all draft models using chain sampling rather than tree-based drafting (Cai et al., 2024) to isolate the contribution of the training objective from inference-time search optimization. In widely used tree-based speculative decoding methods, such as the EAGLE family, tree construction is typically implemented as a heuristic inference-time procedure on top of a pretrained drafter rather than through a different drafter training objective. Since LK losses directly optimize per-position acceptance rates, improvements are expected to transfer to any verification scheme that relies on these acceptance probabilities.

### 5.5. Metrics

We report the expected number of tokens generated per speculation round, computed as $\tau = K \times \frac{\text{\# accepted tokens}}{\text{\# drafted tokens}} + 1$, where $K$ is the maximum draft length. Following standard convention (Leviathan et al., 2023), $\tau$ includes the bonus token that is always sampled from the adjusted target distribution after verification, ensuring at least one token is generated per round. This metric is the major driver of the speedup factor of speculative decoding and serves as our primary evaluation criterion.

We evaluate EAGLE-3 and DeepSeek-MTP with $K = 7$ draft tokens, being consistent with the original EAGLE-3 training and evaluation setup. MEDUSA and MLP speculator are evaluated with $K = 6$ since weights are not shared between decoding heads in these architectures and generation cannot be extended to longer drafts. We demonstrate in Figure 1 that LK losses improve $\tau$ over all values of $K$.

Additionally we report wall-clock speedup measured as the ratio of tokens per second between speculative and vanilla autoregressive decoding under the same target model and hardware configuration in Appendix F.

*Table 1.* Average acceptance length $\tau$ for LLaMA-3.1-8B-Instruct with EAGLE-3, MEDUSA, and MLP draft models.

| Method | Loss | MT-Bench $\tau$ | HumanEval $\tau$ | GSM8K $\tau$ |
|---|---|---|---|---|
| | | Temperature=0 | | |
| EAGLE-3 | KL | 3.75 | 4.82 | 4.50 |
| | TV | 2.81 | 3.42 | 3.34 |
| | $\mathcal{L}_{\text{LK}}^{\alpha}$ | 3.77 | 4.82 | 4.55 |
| | $\mathcal{L}_{\text{LK}}^{\lambda}, \lambda = 0.5$ | 3.78 | 4.84 | 4.53 |
| | $\mathcal{L}_{\text{LK}}^{\lambda}, \eta = 0.7$ | 3.79 | 4.88 | 4.54 |
| | $\mathcal{L}_{\text{LK}}^{\lambda}, \eta = 1$ | 3.80 | 4.83 | 4.54 |
| | $\mathcal{L}_{\text{LK}}^{\lambda}, \eta = 3$ | **3.84** | **4.89** | **4.57** |
| | $\mathcal{L}_{\text{LK}}^{\lambda}, \eta = 10$ | 3.67 | 4.85 | 4.53 |
| MEDUSA | KL | 2.05 | 2.41 | 2.11 |
| | $\mathcal{L}_{\text{LK}}^{\alpha}$ | 2.06 | 2.42 | 2.11 |
| | $\mathcal{L}_{\text{LK}}^{\lambda}, \eta = 10$ | **2.07** | **2.44** | **2.13** |
| MLP | KL | 2.45 | 2.42 | 2.42 |
| | $\mathcal{L}_{\text{LK}}^{\alpha}$ | 2.48 | 2.46 | 2.46 |
| | $\mathcal{L}_{\text{LK}}^{\lambda}, \eta = 3$ | **2.48** | **2.83** | **2.46** |
| | | Temperature=1 | | |
| EAGLE-3 | KL | 3.39 | 4.31 | 3.88 |
| | TV | 2.67 | 3.25 | 3.12 |
| | $\mathcal{L}_{\text{LK}}^{\alpha}$ | 3.50 | 4.48 | 3.98 |
| | $\mathcal{L}_{\text{LK}}^{\lambda}, \lambda = 0.5$ | 3.35 | 4.36 | 3.95 |
| | $\mathcal{L}_{\text{LK}}^{\lambda}, \eta = 0.7$ | **3.53** | 4.45 | 3.98 |
| | $\mathcal{L}_{\text{LK}}^{\lambda}, \eta = 1$ | 3.51 | 4.47 | 3.96 |
| | $\mathcal{L}_{\text{LK}}^{\lambda}, \eta = 3$ | 3.48 | **4.52** | 4.02 |
| | $\mathcal{L}_{\text{LK}}^{\lambda}, \eta = 10$ | 3.34 | 4.51 | **4.03** |
| MEDUSA | KL | 1.72 | 2.02 | 1.81 |
| | $\mathcal{L}_{\text{LK}}^{\alpha}$ | 1.78 | 2.09 | 1.85 |
| | $\mathcal{L}_{\text{LK}}^{\lambda}, \eta = 10$ | **1.85** | **2.22** | **1.92** |
| MLP | KL | 2.13 | 2.16 | 2.16 |
| | $\mathcal{L}_{\text{LK}}^{\alpha}$ | 2.17 | 2.19 | **2.19** |
| | $\mathcal{L}_{\text{LK}}^{\lambda}, \eta = 3$ | **2.19** | **2.62** | 2.18 |

## 6. Evaluation Results

We evaluate LK losses across all target models and draft architectures described in Section 5 with the average acceptance length $\tau$ as the primary metric. See Appendix F for a detailed comparison between objectives and against public HuggingFace checkpoints.

### 6.1. LK Losses across Draft Architectures

Tables 1 presents performance of different draft model architectures trained with various objectives for LLaMA-3.1-8B. The KL baseline demonstrates strong results, but both types of LK losses improve over it across all configurations and sampling temperatures.

Among LK configurations, the hybrid objective with adaptive scheduler achieves the highest acceptance lengths. The likelihood-based objective also outperforms the KL baseline

*Table 2.* Average acceptance length $\tau$ for other target models under different speculative decoding methods. Values in parentheses denote relative improvement (%) of $\mathcal{L}_{LK}^{\lambda}$ ($\eta = 3$) over KL in average acceptance length. MT: MT-Bench; HE: HumanEval; GSM: GSM8K.

| Model | Method/Loss | Temperature = 0 | | | | Temperature = 1 | | | |
|---|---|---|---|---|---|---|---|---|---|
| | | MT ($\tau$) | HE ($\tau$) | GSM ($\tau$) | Mean ($\Delta$%) | MT ($\tau$) | HE ($\tau$) | GSM ($\tau$) | Mean ($\Delta$%) |
| LLaMA 3.1 | EAGLE-3 KL | 3.75 | 4.82 | 4.50 | 4.36 | 3.39 | 4.31 | 3.88 | 3.86 |
| 8B Instruct | EAGLE-3 $\mathcal{L}_{LK}^{\lambda}$ | **3.84** | **4.89** | **4.57** | **4.43 (+1.6)** | **3.48** | **4.52** | **4.02** | **4.01 (+3.9)** |
| LLaMA 3.3 | EAGLE-3 KL | **4.01** | 5.18 | 5.16 | 4.78 | 3.76 | 4.86 | 4.89 | 4.50 |
| 70B Instruct | EAGLE-3 $\mathcal{L}_{LK}^{\lambda}$ | 4.00 | **5.21** | **5.21** | **4.81 (+0.5)** | **3.89** | **5.08** | **5.01** | **4.66 (+3.5)** |
| GPT-OSS 20B | EAGLE-3 KL | 3.31 | 3.07 | **4.01** | 3.46 | 3.12 | 2.89 | 3.51 | 3.17 |
| | EAGLE-3 $\mathcal{L}_{LK}^{\lambda}$ | **3.35** | **3.11** | 4.00 | **3.49 (+0.9)** | **3.20** | **3.01** | **3.65** | **3.29 (+3.8)** |
| GPT-OSS 120B | EAGLE-3 KL | 2.81 | 2.47 | 3.00 | 2.76 | 2.53 | 2.27 | 2.57 | 2.46 |
| | EAGLE-3 $\mathcal{L}_{LK}^{\lambda}$ | **2.86** | **2.51** | **3.06** | **2.81 (+1.8)** | **2.69** | **2.44** | **2.81** | **2.65 (+7.7)** |
| Qwen3-235B-A22B- | EAGLE-3 KL | 3.33 | 4.65 | 4.76 | 4.25 | 2.96 | 4.09 | 4.27 | 3.77 |
| Instruct-2507 | EAGLE-3 $\mathcal{L}_{LK}^{\lambda}$ | **3.36** | **4.74** | **4.87** | **4.32 (+1.8)** | **3.18** | **4.42** | **4.65** | **4.08 (+8.2)** |
| DeepSeek-V3-0324 | MTP original | 2.90 | 3.28 | 3.41 | 3.20 | 2.82 | 3.17 | 3.27 | 3.09 |
| (685B) | MTP KL | 3.96 | 4.74 | 5.67 | 4.79 | 3.66 | 4.39 | 5.23 | 4.43 |
| | MTP $\mathcal{L}_{LK}^{\lambda}$ | **4.00** | **4.77** | **5.72** | **4.83 (+0.8)** | **3.88** | **4.64** | **5.51** | **4.68 (+5.6)** |

though with a smaller margin, particularly under greedy sampling. Yet if the scheduler decay is not optimal ($\eta = 1$), the difference between the two LK losses narrows considerably.

Another important observation is that constant weights in the hybrid objective ($\lambda = 0.5$) make it inferior to any other LK setting. The advantage of the hybrid loss vs KL loss almost disappears, which confirms that curriculum behavior is necessary for effective training.

Training with pure TV loss performs substantially worse than all other objectives. As derived in Section 4.1, TV gradients have severe optimization difficulties when the draft distribution is far from the target. While TV training converges to a meaningful solution, the resulting acceptance lengths are far from being competitive. The hybrid approach addresses this pathology by using KL gradients to guide optimization to the trust region.

A striking pattern emerges when comparing architectures of varying capacity. MEDUSA and MLP speculators with stochastic sampling show average improvements of 7.8% and 8.3% respectively across domains whereas EAGLE-3 sees 3.8% improvement on average. This aligns with our theoretical analysis: low-capacity draft models benefit more from direct optimization of acceptance rate.

### 6.2. Scalability across Target Model Sizes

Table 2 compares performance of our best setting, the hybrid LK loss with $\eta = 3$, against the baseline KL objective across target models ranging from 8B to 685B parameter. Our approach provides consistent improvement over KL, regardless of target model architecture and size.

All EAGLE-3 models in this experiment consist of a single dense transformer layer, while target models range from 32 layers (Llama-3.1-8B) to 94 layers (Qwen3-235B). The most significant improvements occur when targeting large MoE models with much smaller dense draft architectures. GPT-OSS 120B shows +7.7% improvement (compared to +3.8% for GPT-OSS 20B) whilst Qwen3-235B achieves the largest gain of +8.2%. We hypothesize that the large relative difference in parameters and the architectural mismatch create capacity gaps that are difficult to overcome via KL divergence alone.

Another remarkable result is achieved for DeepSeek-V3 with stochastic sampling. As it was stated in Section 5.2, released MTP was not trained primarily for predicting multiple tokens ahead. Fine-tuning it with KL loss substantially improves its performance, but LK loss pushes this bound even further with extra 5.6% gain. This result confirms that our approach is superior to KL not only when the draft distribution starts from random, but generally when the discrepancy between the draft and the target is high.

## 7. Conclusion

A standard practice of training draft models for speculative decoding is minimizing KL divergence. It has the same global optimum as acceptance rate, which typically cannot be achieved with capacity-limited draft models. Our approach addresses this gap through LK losses, training objectives that directly target acceptance rate.

We demonstrate through a relevant example that optimizing TV distance provides stronger results than any of the proxy objectives. Our theoretical analysis reveals caveats in gradient-based TV minimization and establishes a solid

ground for formulating objectives that work in practice.

Extensive experiments across six target models, ranging in size from 8B to 685B parameters, and four draft model architectures demonstrate that LK losses consistently improve acceptance metrics. We observe gains up to $10\%$ in average acceptance length across various task domains, with larger improvements for low-capacity architectures. Our approach introduces no computational overhead during training and integrates directly into existing speculator training pipelines as a drop-in replacement for standard objectives.

**Limitations and future work.** Our experiments demonstrate consistent gains from LK losses across target models and draft architectures, and several directions remain open for further exploration of acceptance-oriented training. The present study focuses on a specific adaptive scheduler for the hybrid objective and uses a fixed exponential aggregation scheme across draft heads. Future work should explore alternative scheduler parameterizations, learnable or data-dependent per-head aggregation strategies, as well as the sensitivity of the adaptive schedule to the order of aggregation. Ablating these alternatives would further clarify the internal mechanisms behind the hybrid objective.

Our evaluation isolates the effect of the training objective under chain sampling and the standard sequential acceptance logic. While improvements in per-position acceptance rates are expected to transfer to more complex inference schemes, systematically verifying this is a natural next step. Evaluating LK-trained speculators under different draft construction strategies, such as tree-based sampling, and under alternative verification or acceptance mechanisms, such as block verification, would broaden the picture.

It would also be valuable to study the trade-off between training-data diversity and epoch count. Our setup uses a large generated corpus with a fixed training schedule, but the relative benefits of broader data coverage versus repeated optimization over fewer samples are an open question. Finally, direct optimization of the draft acceptance length $\tau$, is another promising direction for aligning the objective even more closely with practical speculative decoding speedups.

## Impact Statement

This paper presents work whose goal is to advance the field of Machine Learning, specifically improving the computational efficiency of large language model inference through better training objectives for speculative decoding. There are many potential societal consequences of our work, none of which we feel must be specifically highlighted here.

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

## A. Gradient Derivations

This appendix provides complete derivations of the gradients presented in Section 4. Throughout, we consider distributions $p$ (target, fixed) and $q = \text{softmax}(z_q)$ (draft), where we optimize the logits $z_q$.

### A.1. Softmax Jacobian

The fundamental building block is the Jacobian of the softmax function:

$$\frac{\partial q_i}{\partial z_{q,j}} = q_i(\delta_{ij} - q_j),$$

where $\delta_{ij}$ is the Kronecker delta.

### A.2. KL Divergence Gradient

The forward KL divergence is

$$\text{KL}(p\|q) = \sum_i p_i \log \frac{p_i}{q_i} = \sum_i p_i \log p_i - \sum_i p_i \log q_i.$$

Only the second term depends on $z_q$. Taking derivatives:

$$\begin{aligned}
\frac{\partial\,\text{KL}}{\partial z_{q,j}} &= -\sum_i \frac{p_i}{q_i} \cdot \frac{\partial q_i}{\partial z_{q,j}} \\
&= -\sum_i \frac{p_i}{q_i} \cdot q_i(\delta_{ij} - q_j) \\
&= -\sum_i p_i(\delta_{ij} - q_j) \\
&= -p_j + q_j \sum_i p_i \\
&= q_j - p_j.
\end{aligned}$$

Thus,

$$\boxed{\nabla_{z_q} \text{KL}(p\|q) = q - p.}$$

### A.3. Total Variation Distance Gradient

The TV distance is

$$\text{TV}(p,q) = \frac{1}{2} \sum_i |p_i - q_i|.$$

Define $s_i = \text{sign}(q_i - p_i)$, then $\frac{\partial |p_i - q_i|}{\partial q_i} = s_i$, and:

$$\begin{aligned}
\frac{\partial\,\text{TV}}{\partial z_{q,j}} &= \frac{1}{2} \sum_i s_i \cdot q_i(\delta_{ij} - q_j) \\
&= \frac{1}{2}\left( s_j q_j - q_j \sum_i s_i q_i \right) \\
&= \frac{1}{2} q_j \left( s_j - \mathbb{E}_q[s] \right).
\end{aligned}$$

Thus,

$$\boxed{\nabla_{z_q} \text{TV}(p,q) = \frac{1}{2} q \odot (s - \mathbb{E}_q[s]),}$$

where $\odot$ denotes elementwise multiplication.

### A.4. Negative Log Acceptance Rate Gradient

For the loss $\mathcal{L}_{\mathrm{LK}}^{\alpha} = -\log \alpha$:

$$\frac{\partial \mathcal{L}_{\mathrm{LK}}^{\alpha}}{\partial z_{q,j}} = -\frac{1}{\alpha} \frac{\partial \alpha}{\partial z_{q,j}}$$

$$= \frac{1}{\alpha} \frac{\partial \, \mathrm{TV}}{\partial z_{q,j}}$$

or in vector form:

$$\boxed{\nabla_{z_q} \mathcal{L}_{\mathrm{LK}}^{\alpha} = \frac{1}{\alpha} \nabla_{z_q} \, \mathrm{TV}(p, q).}$$

This key relationship shows that optimizing $-\log \alpha$ is equivalent to optimizing TV with an adaptive learning rate that scales inversely with acceptance rate.

### A.5. Gradient Magnitude Analysis

To understand gradient behavior in practice, we analyze a representative regime that captures early-stage draft model training. Let $V$ denote the size of vocabulary $\mathcal{V}$ and let $S \subset \mathcal{V}$ denote the *support set* – the tokens with non-negligible probability under $p$. Consider a draft distribution $q$ that is approximately uniform ($q_i \approx 1/V$), representing a randomly initialized model, while the target $p$ is concentrated on $|S| = k \ll V$ tokens ($p_i \approx 1/k$ for $i \in S$, and $p_i \approx 0$ otherwise). This regime is relevant because target LLM distributions are typically peaked on few plausible tokens, while undertrained drafts spread mass across the vocabulary.

**KL gradient magnitude.** For $i \in S$: $(q - p)_i \approx 1/V - 1/k \approx -1/k$. For $i \notin S$: $(q - p)_i \approx 1/V$.

$$\|q - p\|^2 \approx k \cdot \frac{1}{k^2} + V \cdot \frac{1}{V^2} = \frac{1}{k} + \frac{1}{V} \approx \frac{1}{k}.$$

Thus $\|\nabla \, \mathrm{KL}\| = \mathcal{O}(1/\sqrt{k})$.

**TV gradient magnitude.** In this regime, $s_i = -1$ for $i \in S$ (since $q_i < p_i$) and $s_i = +1$ for $i \notin S$. Thus $\mathbb{E}_q[s] \approx 1 - 2k/V$. For $i \in S$: $(s_i - \mathbb{E}_q[s])q_i \approx -2/V$. For $i \notin S$: $(s_i - \mathbb{E}_q[s])q_i \approx 2k/V^2 \approx 0$.

$$\|\nabla \, \mathrm{TV}\|^2 \approx \frac{1}{4} \cdot k \cdot \frac{4}{V^2} = \frac{k}{V^2}.$$

Thus $\|\nabla \, \mathrm{TV}\| = \mathcal{O}(\sqrt{k}/V)$, which vanishes for large $V$.

$\mathcal{L}_{\mathrm{LK}}^{\alpha}$ **gradient magnitude.** In this regime, $\alpha \approx k/V$, so:

$$\|\nabla \mathcal{L}_{\mathrm{LK}}^{\alpha}\| = \frac{1}{\alpha} \|\nabla \, \mathrm{TV}\| \approx \frac{V}{k} \cdot \frac{\sqrt{k}}{V} = \frac{1}{\sqrt{k}}.$$

The $1/\alpha$ factor resolves TV's vanishing gradient problem, restoring $\mathcal{O}(1/\sqrt{k})$ magnitude in the diffuse-$q$ regime, while directly targeting acceptance rate.

Table 3 summarizes the gradient components in each region.

*Table 3.* Gradient components for different losses in the diffuse-$q$, concentrated-$p$ regime.

| Loss | Gradient on $S$ | Gradient off $S$ |
|---|---|---|
| KL | $-1/k$ | $+1/V$ |
| TV | $-1/V$ | $\approx 0$ |
| $\mathcal{L}_{\mathrm{LK}}^{\alpha}$ | $-1/k$ | $+1/V$ |

## B. Connection to Negative Log-Likelihood

We establish a relationship between $\mathcal{L}_{\text{LK}}^{\alpha}$ and the standard negative log-likelihood (NLL) used in language model training.

When the target distribution $p$ is a point mass at token $x^*$, i.e., $p(x^*) = 1$ and $p(x) = 0$ for $x \neq x^*$, the acceptance rate simplifies to:

$$\alpha = \sum_i \min(p_i, q_i) = \min(1, q(x^*)) = q(x^*).$$

Therefore:

$$\mathcal{L}_{\text{LK}}^{\alpha}(p, q) = -\log q(x^*),$$

which is precisely the negative log-likelihood of $x^*$ under $q$.

## C. Acceptance Rate as Densities Overlap

Indeed, if we generalize (1) to continuous distributions we get

$$\alpha = \mathbb{E}_{x \sim q} \min\left(1, \frac{p(x)}{q(x)}\right) = \int_{-\infty}^{\infty} \min(q(x), p(x)) dx,$$

which is exactly the total area under the minimum of both density curves.

## D. Rejection Sampling with Greedy Draft Tokens

The standard speculative decoding algorithm (Leviathan et al., 2023) samples draft tokens from the draft distribution $q$ and accepts them with probability $\min(1, p(x)/q(x))$, where $p$ is the target distribution. At non-zero temperatures, proper rejection sampling requires both the numerator $p(x)$ and denominator $q(x)$ to reflect the actual sampling distributions. However, the current vLLM implementation samples draft tokens greedily while still using temperature-scaled target logits in the acceptance criterion.

Under greedy sampling, the draft always selects $x^* = \arg\max_x q(x)$, substituting $q(x^*) = 1$ in the acceptance criterion. The acceptance probability becomes:

$$\alpha_{\text{greedy}} = \min\left(1, \frac{p(x^*)}{1}\right) = p(x^*).$$

When the target distribution is confident and agrees with the draft (high $p(x^*)$), this works well. However, when the target distribution is diffuse, or we are in high-temperature sampling scenario, $p(x^*)$ is small even if the draft correctly identifies the most likely token, leading to systematically low acceptance rates.

Since our LK losses directly optimize the true acceptance rate $\alpha = \sum_x \min(p(x), q(x))$ under temperature $= 1$ evaluating with greedy draft sampling introduces a mismatch between training and evaluation objectives. Our vLLM patch ensures that evaluation faithfully measures the quantity we optimize during training.

## E. Draft Model Architecture Details

For dense target models (Llama-3.1-8B, Llama-3.3-70B), EAGLE-3 draft heads consist of a single transformer layer that mirrors the target model's architecture and processes the concatenation of token embeddings and aggregated hidden states from the target model's intermediate layers. For MoE target models (gpt-oss-20b, gpt-oss-120b, Qwen3-235B), we use a single *dense* transformer block rather than an MoE block. The intermediate dimension of the feed-forward network is chosen as

$$d_{\text{ffn}} = \texttt{num\_experts\_per\_tok} \times d_{\text{expert}},$$

where `num_experts_per_tok` is the number of experts activated per token and $d_{\text{expert}}$ is the intermediate dimension of each expert's FFN. For DeepSeek-V3, we fine-tune the native Multi-Token Prediction (MTP) module, maintaining its original architecture.

MLP Speculator and MEDUSA use simpler architectures: both employ an MLP layer for each head. MEDUSA heads predict all positions in parallel from the same hidden state without token-level autoregression. Unlike EAGLE-3, which shares weights across positions, MLP speculator and MEDUSA train fully independent heads for each speculative position.

# F. Full Experimental Results

*Table 4.* Average acceptance length $\tau$ and end-to-end speedup relative to the baseline without speculative decoding. We evaluate speculative decoding methods trained with different objectives on MT-Bench, HumanEval, and GSM8K in a low-latency setting with batch size 1, and compare them against public Hugging Face checkpoints at temperatures $T=0$ and $T=1$. *Type: Ours* = models trained from scratch with the specified objective; *HF* = public HuggingFace checkpoints evaluated with the same inference pipeline.[6]

| Model | Method | Type | Setup | Temperature = 0 | | | Temperature = 1 | | |
|---|---|---|---|---|---|---|---|---|---|
| | | | | MT-Bench $\tau$ / speedup | HumanEval $\tau$ / speedup | GSM8K $\tau$ / speedup | MT-Bench $\tau$ / speedup | HumanEval $\tau$ / speedup | GSM8K $\tau$ / speedup |
| LLaMA 3.1 8B Instruct | EAGLE-3 | Ours | KL | 3.75 / 2.60 | 4.82 / 3.30 | 4.50 / 3.05 | 3.39 / 2.29 | 4.31 / 3.00 | 3.88 / 2.63 |
| | | | TV | 2.81 / 1.96 | 3.42 / 2.39 | 3.34 / 2.28 | 2.67 / 1.83 | 3.25 / 2.26 | 3.12 / 2.11 |
| | | | $\mathcal{L}_{\text{LK}}^{\alpha}$ | 3.77 / 2.61 | 4.82 / 3.28 | 4.55 / 3.07 | 3.50 / 2.30 | 4.48 / 3.03 | 3.98 / 2.72 |
| | | | $\mathcal{L}_{\text{LK}}, \lambda = 0.5$ | 3.78 / 2.63 | 4.84 / 3.32 | 4.53 / 3.07 | 3.35 / 2.36 | 4.36 / 3.03 | 3.95 / 2.68 |
| | | | $\mathcal{L}_{\text{LK}}^{\lambda}, \eta = 0.7$ | 3.79 / 2.61 | 4.88 / 3.31 | 4.54 / 3.07 | **3.53 / 2.37** | 4.45 / 2.98 | 3.98 / 2.66 |
| | | | $\mathcal{L}_{\text{LK}}^{\lambda}, \eta = 1$ | 3.80 / 2.64 | 4.83 / 3.33 | 4.54 / 3.08 | 3.51 / 2.30 | 4.47 / 3.01 | 3.96 / 2.70 |
| | | | $\mathcal{L}_{\text{LK}}^{\lambda}, \eta = 3$ | **3.84 / 2.62** | **4.89 / 3.35** | **4.57 / 3.10** | 3.48 / 2.39 | **4.52 / 3.08** | 4.02 / 2.72 |
| | | | $\mathcal{L}_{\text{LK}}^{\lambda}, \eta = 10$ | 3.67 / 2.53 | 4.85 / 3.33 | 4.53 / 3.06 | 3.34 / 2.30 | 4.51 / 3.01 | **4.03 / 2.72** |
| | | HF | RH-8B | 3.08 / 2.09 | 3.90 / 2.65 | 3.57 / 2.37 | 2.67 / 1.83 | 3.36 / 2.31 | 3.08 / 2.05 |
| | | | YH-8B | 3.44 / 2.42 | 4.37 / 3.07 | 3.84 / 2.65 | 2.97 / 2.22 | 3.75 / 2.50 | 3.21 / 2.14 |
| | | | ZK-8B | 3.54 / 2.45 | 4.49 / 3.08 | 3.97 / 2.71 | 3.06 / 2.68 | 3.96 / 2.82 | 3.36 / 2.28 |
| LLaMA 3.3 70B Instruct | EAGLE-3 | Ours | KL | **4.01 / 3.01** | 5.18 / 3.87 | 5.16 / 3.78 | 3.76 / 2.82 | 4.86 / 3.66 | 4.89 / 3.61 |
| | | | $\mathcal{L}_{\text{LK}}^{\alpha}$ | 3.95 / 3.00 | 5.12 / 3.82 | 5.13 / 3.78 | 3.76 / 2.85 | 4.94 / 3.71 | 4.91 / 3.66 |
| | | | $\mathcal{L}_{\text{LK}}^{\lambda}, \eta = 3$ | 4.00 / 2.98 | **5.21 / 3.87** | **5.21 / 3.83** | **3.89 / 2.89** | **5.08 / 3.77** | **5.01 / 3.69** |
| | | HF | RH-70B | 3.11 / 2.35 | 3.99 / 2.99 | 3.62 / 2.68 | 2.88 / 2.10 | 3.62 / 2.69 | 3.29 / 2.44 |
| | | | YH-70B | 3.13 / 2.50 | 3.96 / 3.12 | 3.76 / 2.90 | 2.77 / 2.07 | 3.49 / 2.63 | 3.34 / 2.50 |
| GPT-OSS 20B | EAGLE-3 | Ours | KL | 3.31 / 1.41 | 3.08 / 1.32 | 4.01 / 1.63 | 3.12 / 1.30 | 2.89 / 1.20 | 3.51 / 1.44 |
| | | | $\mathcal{L}_{\text{LK}}^{\alpha}$ | 3.27 / 1.41 | 3.07 / 1.32 | **4.03 / 1.63** | **3.28 / 1.35** | 2.97 / 1.25 | 3.65 / 1.49 |
| | | | $\mathcal{L}_{\text{LK}}^{\lambda}, \eta = 3$ | **3.35 / 1.42** | **3.11 / 1.34** | 4.00 / 1.63 | 3.20 / 1.35 | **3.01 / 1.26** | **3.65 / 1.49** |
| | | HF | RH-20B[†] | 2.90 / 1.26 | 2.70 / 1.17 | 3.49 / 1.45 | 2.63 / 1.12 | 2.43 / 1.04 | 3.00 / 1.26 |
| GPT-OSS 120B | EAGLE-3 | Ours | KL | 2.81 / 1.57 | 2.47 / 1.54 | 3.00 / 1.77 | 2.53 / 1.20 | 2.27 / 1.19 | 2.57 / 1.32 |
| | | | $\mathcal{L}_{\text{LK}}^{\alpha}$ | 2.80 / 1.59 | 2.51 / 1.57 | **3.08 / 1.82** | 2.67 / 1.49 | 2.39 / 1.45 | 2.78 / 1.63 |
| | | | $\mathcal{L}_{\text{LK}}^{\lambda}, \eta = 3$ | **2.86 / 1.62** | **2.51 / 1.60** | 3.06 / 1.82 | **2.69 / 1.48** | **2.44 / 1.48** | **2.81 / 1.66** |
| Qwen 3 235B A22B Instruct | EAGLE-3 | Ours | KL | 3.33 / 2.09 | 4.65 / 2.73 | 4.76 / 2.88 | 2.96 / 1.84 | 4.09 / 2.37 | 4.27 / 2.50 |
| | | | $\mathcal{L}_{\text{LK}}^{\alpha}$ | 3.29 / 2.10 | 4.68 / 2.77 | 4.82 / 2.91 | 3.11 / 1.96 | 4.31 / 2.46 | 4.50 / 2.61 |
| | | | $\mathcal{L}_{\text{LK}}^{\lambda}, \eta = 3$ | **3.36 / 2.13** | **4.74 / 2.78** | 4.87 / 2.93 | **3.18 / 1.98** | **4.42 / 2.52** | **4.65 / 2.68** |
| | | HF | RH-235B | 2.92 / 1.85 | 4.06 / 2.42 | 4.31 / 2.63 | 2.56 / 1.61 | 3.52 / 2.06 | 3.78 / 2.22 |
| | | | ZK-235B[‡] | 3.06 / 1.94 | 4.54 / 2.70 | **4.90 / 2.94** | 2.64 / 1.63 | 4.03 / 2.36 | 4.43 / 2.57 |
| DeepSeek V3 0324 | MTP | Ours | KL | 3.96 / 2.22 | 4.74 / 2.62 | 5.67 / 2.95 | 3.66 / 1.85 | 4.39 / 2.17 | 5.23 / 2.30 |
| | | | $\mathcal{L}_{\text{LK}}^{\lambda}$ | **4.00 / 2.22** | **4.77 / 2.64** | **5.72 / 2.97** | **3.88 / 2.14** | **4.64 / 2.42** | **5.51 / 2.71** |
| | | HF | DS-685B | 2.90 / 1.66 | 3.28 / 1.84 | 3.41 / 1.82 | 2.82 / 1.54 | 3.17 / 1.59 | 3.27 / 1.47 |

[†] Results are based on the checkpoint revision available at the time experiments were conducted, prior to the subsequent weight update released during the review period.
[‡] This checkpoint employs a wider MLP (24,576 vs. 12,288) and a smaller vocabulary (32k vs. 64k), yielding a slightly larger parameter count.

---

[6]RH=RedHatAI, YH=yuhuili, ZK=zhuyksir, DS=deepseek-ai.
RH-8B=Llama-3.1-8B-Instruct-speculator.eagle3; YH-8B=EAGLE3-LLaMA3.1-Instruct-8B;
ZK-8B=EAGLE3-Llama-3.1-8B-Instruct;
RH-70B=Llama-3.3-70B-Instruct-speculator.eagle3; YH-70B=EAGLE3-LLaMA3.3-Instruct-70B;
RH-20B=gpt-oss-20b-speculator.eagle3; RH-235B=Qwen3-235B-A22B-Instruct-2507-speculator.eagle3;
ZK-235B=EAGLE3-Qwen3-235B-A22B-Instruct-2507-FP8; DS-685B=DeepSeek-V3-0324.

