# OpenReview forum: "LK Losses: Direct Acceptance Rate Optimization for Speculative Decoding"
_ICML.cc/2026/Conference — ICML 2026 regular_

### Official Review · Reviewer_FfwQ · 2026-03-11

**Soundness:** 3
**Presentation:** 2
**Significance:** 3
**Originality:** 3
**Overall Recommendation:** 4
**Confidence:** 4

**Summary:**

This paper proposes LK losses, two training objectives for speculative decoding that directly target draft acceptance rather than relying on KL divergence as a proxy. The motivation is that, for capacity-limited draft models, optimizing KL does not necessarily lead to the best acceptance behavior during verification. To address this mismatch, the paper introduces one loss based on negative log-acceptance and a second hybrid objective that gradually transitions from KL-style optimization to more direct acceptance-oriented training. Experiments across four draft architectures and six target models, covering general, code, and math settings, show consistent improvements in average acceptance length over standard KL-based training, with gains of up to about 10%, while preserving inference-time efficiency.

**Compliance With Llm Reviewing Policy:**

Affirmed.

**Final Justification:**

Some of my concerns remain, particularly regarding the sensitivity of the adaptive schedule in the hybrid objective, where the rebuttal does not fully resolve the question about the aggregation order. Nevertheless, the authors addressed most of my other concerns, so I will maintain my score at Weak Accept.

**Key Questions For Authors:**

1.	The paper reports consistent gains in average acceptance length, but the authors should provide end-to-end throughput or wall-clock speedup results under identical hardware and batching.
2.	The evaluation is limited to chain sampling. Since tree-style drafting is central to modern speculative decoding and changes the effective optimization target, it would be important to show whether LK losses remain effective in tree-based inference settings, such as EAGLE-style trees.
3.	Since the hybrid objective relies on an acceptance-driven schedule, how sensitive is the method to the practical estimation and aggregation of $\alpha$?

**Limitations:**

yes

**Strengths And Weaknesses:**

**Strengths**

1.	The paper addresses a well-motivated problem in speculative decoding: KL is only a proxy for draft quality, while the true objective is acceptance during verification. The proposed LK losses directly target this mismatch instead of modifying the drafting architecture itself.
2.	The technical design is clean and reasonably principled. The paper introduces a direct acceptance-oriented objective and an adaptive hybrid objective that blends KL and TV, with the schedule driven by the current acceptance rate. The gradient discussion clearly motivates why pure TV is difficult to optimize and why the hybrid objective is useful.
3.	The empirical study is fairly broad and convincing. The method is tested across multiple draft architectures and target models, and the reported gains in average acceptance length are consistent across general, code, and math settings, under both greedy and stochastic sampling.

**Weaknesses**

1.	The paper focuses on average acceptance length, but does not provide enough end-to-end system measurements such as wall-clock speedup or throughput. Since better acceptance does not always translate linearly into practical decoding speed, this weakens the systems-level significance of the results.
2.	The comparisons are mainly against standard KL training. More direct baselines that also aim to improve acceptance-oriented training or reduce the training-inference mismatch would make the empirical claim stronger, like distillspec [1] or adaspec [2].
3.	The evaluation is limited to chain sampling. Since tree-based verification is the current SOTA and involves non-linear path-finding gains, maximizing per-token acceptance in a chain sequence may not lead to optimal tree configurations. The authors should clarify if the objective requires modification for tree-style drafting.



[1] Distillspec: Improving speculative decoding via knowledge distillation.

[2] Adaspec: Selective knowledge distillation for efficient speculative decoders.

---

> ### Author Rebuttal · Authors · 2026-03-31
>
> Thank you for the positive assessment and for raising two important questions: comparison to related approaches based on Knowledge Distillation (KD) and the scope of the paper with respect to tree-based drafting.
>
> As for the DistillSpec, AdaSpec and related methods, we agree that these are important adjacent works and our paper should position itself more clearly with respect to them. At the same time, they are not direct drop-in replacements in our setting. DistillSpec and AdaSpec primarily study KD recipes for standalone, typically pre-trained language models being used as drafters, focusing on such choices as dataset construction, divergence loss selection and better utilization of the model capacity. Our setting is different as we study native draft modules tightly integrated with target models and trained in one stage from scratch. One of the main research questions in our work is whether the core training objective itself should directly optimize acceptance rather than a proxy divergence measure. This is why forward KL is our main baseline, and why we also explore direct TV optimization. We will make this distinction more explicit in the related work and experimental discussion in the next paper revision.
>
> Regarding the question about tree-based drafting, we agree that there is highly non-trivial relation between chain- and tree-sampling performance, but which nevertheless demonstrates a strongly positive correlation in practice. This is exactly why we kept tree sampling out of scope and focused purely on chain setting: to isolate the effect of the training objective from extra inference-time search optimizations. In typical tree verification being widely adopted in modern speculators, particularly in EAGLE family, tree construction is a heuristic-based inference-time procedure built on top of a pre-trained drafter, which does not define a different training objective for the draft model. Learned RL-based tree builders do introduce an additional optimization problem, but that is a separate research direction from the one studied here. In our paper we stay within the single-chain teacher-forced setting and investigate whether acceptance-centric training improves the underlying drafter. We will clarify this scope more explicitly in the limitations in the next paper revision.
>
> As for end-to-end speedup, please see our response to [Reviewer GTTW](https://openreview.net/forum?id=ZqnVidXtxV&noteId=YGRsHDnD5f). We agree that these numbers are extremely important for drawing a full picture and are actively working on that.
>
> Regarding the sensitivity of the adaptive schedule in the hybrid objective, we agree that ablation of the aggregation order (i.e. first aggregate and then compute weights or vice versa) would be helpful for better understanding its internal mechanisms. We will add it to the future work.
>
> **NOTE**. After the submission deadline, we discovered a small bug in our evaluation configuration for some draft models. This bug changes slightly absolute values of average acceptance lengths, but keeps relative gains of LK loss over KL consistent with our reported findings. For details, please see our response to [Reviewer Ekpt](https://openreview.net/forum?id=ZqnVidXtxV&noteId=UEwwafQmze).

---

> > ### Author Rebuttal · Reviewer_FfwQ · 2026-04-02
> >
> > I will maintain the score for weak acceptance.

---

### Official Review · Reviewer_6jtm · 2026-03-13

**Soundness:** 3
**Presentation:** 4
**Significance:** 4
**Originality:** 4
**Overall Recommendation:** 6
**Confidence:** 4

**Summary:**

In order to train draft models in speculative decoding, the KL divergence is minimised, as it theoretically leads to the same global optimum as maximising the acceptance rate (or equivalently, minimising the total variation TV distance). In practice however, this might be suboptimal due to the difficulty of reaching the global optimum. Indeed, this would require the limited-capacity draft model to capture well the distribution of a potentially much higher-capacity target model. The authors illustrate this with an example showing that optimising the TV distance, however, leads to better acceptance metrics. In order to circumvent theoretical limitations of the TV distance, they propose LK losses, which are constructed from the KL divergence and the TV distance. They construct various types of drafters using existing baselines (EAGLE, MTP, …), comparing the use of KL and LK as drafter training losses. The proposed LK loss is shown to produce draft models with improved acceptance rates over drafters trained with just the KL divergence, with stronger effects when sampling stochastically drafters of larger target models.

**Compliance With Llm Reviewing Policy:**

Affirmed.

**Final Justification:**

Like I said, I am quite happy with this paper and the authors promised to address the easily-addressable weaknesses I mentioned, so I raise my score to a 6 in anticipation of those changes.

@AC, I think this paper deserves to be flagged for a spotlight.

**Key Questions For Authors:**

**Questions**
1. p5: “EAGLE-3 uses a truncated vocabulary for its LM head as it is proposed in FR-Spec” I don’t understand; I don’t see that being mentioned in the EAGLE-3 paper, which was released a dozen days after FR-Spec on arXiv. I also couldn’t find FR-Spec support in vLLM. I’m very confused; can you explain what you mean here?
2. p6: “using only the vocabulary definitions while training all the weights from scratch.” what do you mean by that?
3. p6: “This ensures the draft model is trained on the same distribution it will encounter during inference.” are you referring to the fact that you’re training on synthetic data from the target model to ensure that you stay “on-policy” with the target model? If so, “All draft models except DeepSeek-MTP are trained from scratch for 10 epochs.” generally, such a degree of repeats tends to hurt when training an LLM (here the drafter) from scratch; would it not be better to reduce the number of epoch by instead growing your synthetic data (e.g. 2 epochs but 5 generations of the target model per base prompt)?
4. p6: “All the losses are aggregated across draft heads with exponential weight decay” makes me wonder (I don’t expect you to run this): wouldn’t modifying $\eta$ based on the head position help?


**Suggestions**
1. p7: “We evaluate all draft models using chain sampling rather than tree-based drafting” I recommend briefly explaining those concepts so readers unfamiliar with them may understand your argument (perhaps in the appendix if you want), especially since your paper is quite accessible overall.
2. p8: “All EAGLE-3 models in this experiment consist of a single dense transformer layer, while target models range from [...]” add somewhere in this paragraph that your analysis/observations are made for stochastic sampling, since you don’t necessarily get a clear trend of drafters of larger target models benefitting more from optimising the acceptance rate than those of smaller target models.


**Typos**

Congratulations, I didn’t see any.

**Limitations:**

No, see weaknesses.

**Strengths And Weaknesses:**

**Strengths**

I am quite happy with this paper:
1. It is very well-written and I believe it will be accessible to a wide audience.
2. Efficient inference is arguably one of the most important problems in ML right now, given the disproportionate computational, monetary, energetic and environmental cost of foundation models. Even a few percentages can make a big difference.
3. Their experiments are very promising since the method appears to always lead to an improvement. The methodological setup is convincing due to testing several methods to build drafters and using various sizes and architectures of target models. Their method is orthogonal to (i.e., can be used along with) many existing speculative decoding techniques. It is also very easy to implement, so as usual with such practical methods, people can easily experiment with it.
4. The authors used vLLM for experiments, which makes the setup more realistic for readers wanting results more representative of production settings.


**Weaknesses**

Again, I’m overall convinced this paper deserves acceptance. However, before I can raise my score to a clear accept, please address the following weaknesses:
1. See question 1. It’s very important to clarify that, as it affects the baseline you’re comparing to and any attempt at reproducing your results.
2. Please add further details such as the sequence length used for training or the memory and wall-time cost of your method relative to the baseline during training. If it’s negligible, state it. Add also any version of libraries used (e.g. vLLM) for reproducibility.
3. No discussion of limitations. I see a few, but to give you an idea of what a reader might think reading your paper: “Are those really the best $\eta$ in every setting” or “how would other schedules than the one in eq 5 perform?”. Basically, think of design decisions: do you show an ablation for all of them in the paper? Those are limitations.

**More details about weaknesses**

Nice job, no further detail I have to write about!

---

> ### Author Rebuttal · Authors · 2026-03-31
>
> We thank the reviewer for the encouraging overall assessment and for detailed technical questions and useful suggestions.
>
> We agree that the paper does not frame clearly the limitations of the method, so we will improve on that in the next revision by adding various research questions not being covered by this study. Particularly, it concerns other possible designs of the adaptive scheduler and hybrid loss aggregation as well as evaluating performance under different type of draft construction (e.g. tree sampling) or acceptance logic (e.g. block verification).
>
> We agree that the paper should be more explicit about the training recipe. In the next paper revision we will add a section to the appendix with more details including the training sequence length (8k) and exact vLLM version (v0.11.0). Since our method changes only the loss and not the draft architecture, the extra training cost relative to the standard KL training is negligible and comes mainly from computing acceptance-based weights.
>
> We also agree that the wording around EAGLE-3 and truncated vocabularies might be a bit confusing and will revise it. We did not mean that the EAGLE-3 paper itself involves FR-Spec or similar approach. Our point was only that the speculator setup being used in many practical implementations employs a reduced draft vocabulary, including the [released weights](https://huggingface.co/yuhuili/EAGLE3-LLaMA3.1-Instruct-8B/blob/main/config.json#L23) from the authors of original EAGLE-3 paper. In our experiments, we follow that best practice and import only those vocabulary definitions (draft token ids → target token ids mappings) from open-source checkpoints, while training the model weights themselves from scratch. Thus, we do not reuse pre-trained LM-head weights anyhow.
>
> Regarding the sentence that target-generated synthetic data “ensures the same distribution as inference” - yes, by this we mean that the training data is on-policy with respect to the target model. We agree that generating more diverse synthetic data, for example multiple generations per prompt, is a promising research direction. Our current choice was partly motivated by practical considerations: generating synthetic data from large target models (up to 685B parameters) is already one of the most time-consuming steps in the training pipeline, and scaling to 5× more generations per prompt would proportionally increase this cost. Additionally, we note that draft heads differ from standalone small language models in that they reuse rich hidden features from the target model and primarily optimize short-horizon draft proposal quality rather than standalone language modeling capability. We assume that draft heads are less sensitive to the diversity of full responses than a language model trained from scratch, though we acknowledge this is still a hypothesis rather than a proven fact. We also agree that systematically studying the trade-off between data diversity and epoch count is a valuable direction and will include this to the future work section.
>
> Regarding the aggregation of losses across draft heads, position is already taken into account through exponential head weighting, so later heads receive smaller weight than earlier ones. This is a standard setting in MEDUSA paper and practical implementations of EAGLE-3. We are also exploring more advanced weighting / curriculum approaches, but this is ongoing research that goes beyond the scope of this paper.
>
>
> **NOTE**. After the submission deadline, we discovered a small bug in our evaluation configuration for some draft models. This bug changes slightly absolute values of average acceptance lengths, but keeps relative gains of LK loss over KL consistent with our reported findings. For details, please see our response to [Reviewer Ekpt](https://openreview.net/forum?id=ZqnVidXtxV&noteId=UEwwafQmze).

---

> > ### Author Rebuttal · Reviewer_6jtm · 2026-04-03
> >
> > Thank you for the rebuttal!
> >
> > > Our current choice was partly motivated by practical considerations: generating synthetic data from large target models (up to 685B parameters) is already one of the most time-consuming steps in the training pipeline, and scaling to 5× more generations per prompt would proportionally increase this cost.
> >
> > Perfectly understandable, lost track of the 685B model and had the smaller one in mind when I suggested that.
> >
> > > We assume that draft heads are less sensitive to the diversity of full responses than a language model trained from scratch, though we acknowledge this is still a hypothesis rather than a proven fact. We also agree that systematically studying the trade-off between data diversity and epoch count is a valuable direction and will include this to the future work section.
> >
> > I strongly suspect that less diversity of data for the drafter will translate in higher rejection rates. Probably not a big issue for specialised models, probably more impactful for generalist LLMs serving hundreds of millions.
> >
> > > NOTE. After the submission deadline, we discovered a small bug in our evaluation configuration for some draft models. This bug changes slightly absolute values of average acceptance lengths, but keeps relative gains of LK loss over KL consistent with our reported findings. For details, please see our response to Reviewer Ekpt.
> >
> > Thank you for your honesty. I believe it does not affect the merits of the paper.
> >
> > ___
> >
> > Like I said, I am quite happy with this paper and the authors promised to address the easily-addressable weaknesses I mentioned, so I raise my score to a 6 in anticipation of those changes.

---

### Official Review · Reviewer_Ekpt · 2026-03-13

**Soundness:** 3
**Presentation:** 3
**Significance:** 3
**Originality:** 3
**Overall Recommendation:** 4
**Confidence:** 3

**Summary:**

This paper proposes LK loss, an alternative loss for training draft models for speculative decoding. Instead of using the typical KL divergence objective, the paper reformulates the acceptance rate in speculative sampling scenarios and proposes directly using it as a training objective. The proposed loss consistently improves the acceptance length across diverse configurations, including four draft architectures and six target models ranging from 8B to 685B parameters.

**Compliance With Llm Reviewing Policy:**

Affirmed.

**Final Justification:**

I recommend acceptance due to the reasons listed in the 'strengths and weaknesses' section. The rebuttal also successfully resolved my concerns regarding hyperparameter sensitivity.

**Key Questions For Authors:**

[Q1] How sensitive is the model to different values of the hyperparameter $\eta$?

**Limitations:**

yes

**Strengths And Weaknesses:**

**Strengths**

[S1] The paper is clearly written and easy to follow.

[S2] The proposed loss is simple and effective while being theoretically grounded. The motivation is clear, straightforward, and persuasive.

[S3] The proposed loss shows strong empirical gains, with consistent improvements across diverse evaluation setups. The experiments are thorough and extensive, covering multiple models and sampling configurations.

**Weaknesses**

[W1] The proposed loss appears to be sensitive to hyperparameters. While the manuscript includes experiments with two different $\eta$ values, it would be helpful to include a more thorough analysis of hyperparameter sensitivity.

---

> ### Author Rebuttal · Authors · 2026-03-31
>
> Thank you for the positive assessment and for raising the question about hyperparameter sensitivity of the LK hybrid objective - we agree it is important to clarify.
>
> The main tunable component in LK is the schedule/mixing behavior and the corresponding decay parameter, rather than a large collection of method-specific hyperparameters. Our current evidence suggests that performance of our method is reasonably stable over a practical range of the decay parameter. The benefit of the hybrid LK loss does not appear to depend on exhaustive tuning of this parameter, but rather on presence of the adaptive scheduler. We agree, however, that this should be shown explicitly rather than only stated. We are therefore running additional ablations over the relevant schedule decay values and will add a compact sensitivity analysis to the appendix in the next paper revision.
>
> More precisely, we have obtained the following results for Llama 3.1 8B over the set of different values of $\eta$:
>
> | η            | MTBench (T=0) | HumanEval (T=0) | GSM8K (T=0) | MTBench (T=1) | HumanEval (T=1) | GSM8K (T=1) |
> |--------------|---------------|-----------------|-------------|---------------|-----------------|-------------|
> | 0 (pure KL)  | 3.75          | 4.82            | 4.50        | 3.39          | 4.31            | 3.88        |
> | 0.7          | 3.79          | 4.88            | 4.54        | 3.53          | 4.45            | 3.98        |
> | 1            | 3.80          | 4.83            | 4.54        | 3.51          | 4.47            | 3.96        |
> | 3            | 3.84          | 4.89            | 4.57        | 3.48          | 4.52            | 4.02        |
> | 10           | 3.67          | 4.85            | 4.53        | 3.34          | 4.51            | 4.03        |
>
> This range is driven by the semantics of this hyperparameter which makes sure the TV term in the hybrid loss eventually gets more weight than the KL term. Thus, the following condition should be met: $\exp(-\eta) < 0.5$, i.e. KL weight is less than TV weight when acceptance rate reaches 1. Therefore, the absolute minimum which would make sense is $\eta > \ln 2 \approx 0.69$. On the other hand, we wish to start direct optimization as soon as possible, which suggests higher values of $\eta$ and steeper KL decay curve. However, if the hyperparameter is too high, we may switch to TV too early, while it still has small gradients issue.
>
>
> **NOTE.** After the submission deadline, we discovered a bug in our RoPE configuration for the LLaMA-3.3-70B and Qwen3-235B EAGLE-3 draft models. The bug affected only vLLM evaluation whereas all training was conducted correctly, so the model weights themselves remain unchanged. Since the same pipeline was used for all objectives, the bug impacted KL baselines and LK-trained models equally. After fixing the bug and re-evaluating, the absolute values of acceptance lengths for these two models improved across the board, while the relative gains of LK over KL remain consistent with our reported findings. All other models and all conclusions are unaffected. We will incorporate the corrected numbers into the next paper revision. The updated rows of Table 2 (average acceptance length) are as follows:
> |     Model             |    Loss           | MT (T=0)                 | HE (T=0) | GSM (T=0)| Mean (Δ%)   | MT (T=1)                 | HE (T=1) | GSM (T=1)| Mean (Δ%)   |
> |------------------|---------------|------------------------|--------|--------|-------------|------------------------|--------|--------|-------------|
> | LLaMA 3.3 70B    | KL            | 4.01                   | 5.18   | 5.16   | 4.78        | 3.76                   | 4.86   | 4.89   | 4.50        |
> |                  | $L_{\mathrm{LK}}^\lambda$        | 4.00                   | 5.21   | 5.21   | 4.81 (+0.5) | 3.89                   | 5.08   | 5.01   | 4.66 (+3.5) |
> | Qwen3-235B       | KL            | 3.33                   | 4.65   | 4.76   | 4.25        | 2.96                   | 4.09   | 4.27   | 3.77        |
> |                  | $L_{\mathrm{LK}}^\lambda$        | 3.36                   | 4.74   | 4.87   | 4.32 (+1.8) | 3.18                   | 4.42   | 4.65   | 4.08 (+8.2) |
>
> We apologize for the oversight.

---

> > ### Author Rebuttal · Reviewer_Ekpt · 2026-04-03
> >
> > Thanks for the detailed explanations. I will keep my positive score to recommend acceptance.

---

### Official Review · Reviewer_GTTW · 2026-03-13

**Soundness:** 3
**Presentation:** 3
**Significance:** 2
**Originality:** 3
**Overall Recommendation:** 4
**Confidence:** 5

**Summary:**

This paper proposes LK Loss, a new objective designed to directly optimize the acceptance rate in speculative decoding (SD). The authors observe that most existing SD training methods optimize surrogate objectives such as KL divergence or NLL, which may not align well with the actual objective of speculative decoding: maximizing the acceptance probability between the draft and target models. To address this, the paper derives a loss based on the acceptance probability formulation and proposes a differentiable approximation to optimize it during draft model training.
Experiments are conducted on several datasets to demonstrate improvements in average accepted length over KL-based baselines.
The paper addresses an important problem in LLM inference acceleration. However, I have several concerns regarding the evaluation methodology, experimental setup, and comparisons with existing work, which currently weaken the empirical support for the paper’s claims.

**Compliance With Llm Reviewing Policy:**

Affirmed.

**Final Justification:**

Maintain the positive score.

**Key Questions For Authors:**

1.	Can the authors report actual inference speedup (α) or wall-clock latency improvements in addition to average accepted length?
2.	Why was the modified vLLM decoding pipeline used for evaluation instead of the standard speculative decoding implementation?
3.	How sensitive is the proposed loss to its hyperparameters?
4.	Have the authors compared the proposed loss with alternative objectives proposed in DistillSpec or related SD training methods?

**Limitations:**

yes

**Strengths And Weaknesses:**

Strengths
1.	The paper addresses an important and practical problem in large language model inference: improving the efficiency of speculative decoding. As LLM deployment increasingly faces latency and cost constraints, improving the acceptance rate of draft tokens is a meaningful direction for reducing inference overhead and accelerating generation.
2.	The paper identifies a conceptual mismatch between commonly used training objectives (e.g., KL divergence or NLL) and the actual objective of speculative decoding, which is to maximize the acceptance probability between draft and target models. This observation is well motivated and aligns with the theoretical formulation of speculative decoding.
3.	The proposed LK loss attempts to directly optimize the acceptance probability rather than relying on surrogate objectives. This perspective is intuitive and conceptually appealing, since the acceptance probability directly determines how many draft tokens survive verification.
4.	The method itself is relatively simple and does not require changes to the speculative decoding algorithm at inference time. Instead, it modifies the training objective of the draft model, which makes it potentially easy to integrate into existing speculative decoding training pipelines.
Weaknesses
1.	The evaluation only reports the average accepted length, which does not necessarily translate into real inference speedup. In speculative decoding systems, the actual acceleration depends on multiple factors, including draft generation cost, target verification cost, batching efficiency, and system overhead. Therefore, an increase in accepted length does not automatically imply proportional latency reduction. It would be more convincing if the paper reported system-level metrics such as speedup ratio (α), latency, or tokens-per-second throughput.
2.	The experimental evaluation is conducted on only a limited number of datasets, and the improvements over baselines appear relatively small. Given that speculative decoding behavior can vary significantly across tasks and domains, it would be important to demonstrate consistent gains across a broader range of benchmarks or model configurations. With the current experimental scope and modest improvements, it is difficult to assess whether the proposed loss provides a robust advantage.
3.	The evaluation protocol modifies the speculative decoding implementation in vLLM. In particular, the KL baseline may perform reasonably well under the standard speculative decoding setup in the official vLLM implementation (which uses greedy draft sampling), but in this paper it is evaluated under a modified decoding configuration based on rejection sampling. As a result, the baseline is effectively evaluated under a different setup than the one commonly used in practice. This raises fairness concerns, since the authors are introducing a modified evaluation environment and then comparing methods within that environment.
4.	The proposed method appears to involve several hyperparameters, but the paper provides limited analysis regarding hyperparameter sensitivity. It is unclear how robust the performance improvements are across different parameter settings or training configurations. Without such analysis, it is difficult to determine whether the reported gains are stable or depend on careful tuning.
5.	The paper does not compare the proposed loss with several existing loss formulations that have been explored in prior speculative decoding work (e.g., DistillSpec and related methods). Since these works also investigate alternative objectives beyond standard KL-based training, including them as baselines would provide a more comprehensive evaluation of whether the proposed LK loss offers meaningful advantages.

---

> ### Author Rebuttal · Authors · 2026-03-31
>
> We thank the reviewer for the thorough analysis of and thoughtful comments to our paper. We highly appreciate recognizing the motivation behind our approach and its practical simplicity.
>
> We agree that end-to-end metrics are extremely important, especially the speed-up evaluation. Acceptance length is our main metric because LK changes only the training objective of the drafter, not its architecture, runtime algorithm or inference settings, so it is the most direct quantity affected by the method.
>
> Still, we have now run output throughput (tok/s) measurements for Qwen3-235B-Instruct-2507 under stochastic chain sampling (T=1):
> | Dataset   | Batch size | Baseline (no SD) | KL     | LK-hybrid | Speedup (LK over KL) |
> |-----------|------------|------------------|--------|-----------|----------------------|
> | MTBench   | 1          | 99.11            | 189.30 | 209.70    | +10.8%               |
> |           | 2          | 175.58           | 302.79 | 338.45    | +11.7%               |
> | HumanEval | 1          | 97.41            | 262.57 | 285.97    | +8.9%                |
> |           | 2          | 171.62           | 406.44 | 441.01    | +8.5%                |
> | GSM8K     | 1          | 98.37            | 263.61 | 291.35    | +10.5%               |
> |           | 2          | 173.01           | 424.49 | 465.65    | +9.7%                |
>
> We are extending the same measurements to the remaining models in our study and will add them in the next paper revision.
>
> Regarding the concern that the empirical study may be too narrow, we agree that wider and more diverse validation would make the empirical evidence of the paper more convincing. However, we would like to emphasize that we evaluate on three domains of practical interest - general dialogue, coding and math - and, unlike a number of prior speculative decoding works, we evaluate on the **full** datasets rather than their small subsets.
>
> Regarding the modified vLLM setup, our goal was to make the comparison accurate. As described in Appendix D of the paper, rejection sampling with greedy draft tokens, being supported in vLLM at the time of our experiments, creates a mismatch between the stochastic acceptance rule and the actual drafting process. Because LK directly optimizes acceptance under the true stochastic setting, we patched vLLM to implement proper lossless rejection sampling so that training and evaluation are aligned. Now, a [proper stochastic sampling](https://github.com/vllm-project/vllm/pull/35461) from the draft model has been introduced in vLLM starting from v0.18.0 (released on Mar 20, 2026), so if one uses the most recent version of vLLM no patching is needed.
>
> As for the loss hyperparameter sensitivity question, we agree that the paper should show this more directly and will add ablations. For details, please see our response to [Reviewer Ekpt](https://openreview.net/forum?id=ZqnVidXtxV&noteId=UEwwafQmze).
>
> As for the question on DistillSpec/AdaSpec and related baselines, please see our detailed response to [Reviewer FfwQ](https://openreview.net/forum?id=ZqnVidXtxV&noteId=lddPPRw91l).
>
> **NOTE.** After the submission deadline, we discovered a small bug in our evaluation configuration for some draft models. This bug changes slightly absolute values of average acceptance lengths, but keeps relative gains of LK loss over KL consistent with our reported findings. For details, please see our response to [Reviewer Ekpt](https://openreview.net/forum?id=ZqnVidXtxV&noteId=UEwwafQmze).

---

> > ### Author Rebuttal · Reviewer_GTTW · 2026-04-07
> >
> > Thank you for the thorough rebuttal. My main concerns have been adequately addressed. I will maintain my positive score.

---

### Decision · Program_Chairs · 2026-04-30

**Decision:**

Accept (regular)

**Comment:**

The paper presents LK losses, a novel family of training objectives designed to optimize speculative decoding by directly targeting the token acceptance rate. While traditional methods rely on Kullback-Leibler (KL) divergence as a proxy, the authors argue that for capacity-limited draft models, KL minimization does not necessarily maximize the acceptance rate. They propose a hybrid loss that transitions from KL-style optimization to a direct total variation (TV) distance minimization using an adaptive scheduler. The approach is evaluated across a robust range of architectures (4 draft types, 6 target models) including extremely large scales (up to 685B parameters).

The reviewers reached a consensus that the paper provides a technically sound and practically valuable contribution to LLM inference optimization. Reviewer 6jtm championed the paper for its "exceptional impact" and "flawless" presentation, while others noted the method's simplicity and effectiveness. The authors' proactive rebuttal, particularly the addition of wall-clock speedup metrics and a hyperparameter sensitivity analysis, resolved the primary concerns regarding empirical significance. The AC agrees that the shift toward direct acceptance optimization is a well-grounded and promising direction for the field.